# Exposure to Intimate-Partner Violence and Resilience Trajectories of Adolescents: A Two-Wave Longitudinal Latent Transition Analysis

**DOI:** 10.3390/ijerph20095676

**Published:** 2023-04-28

**Authors:** Dilan Aksoy, Celeste Simões, Céline Anne Favre

**Affiliations:** 1Department of Research and Development, School of Education, University of Applied Sciences and Arts Northwestern Switzerland, 5210 Windisch, Switzerland; 2Faculdade de Motricidade Humana, Universidade de Lisboa, 1495-751 Lisbon, Portugal; 3Instituto de Saúde Ambiental, Faculdade de Medicina, Universidade de Lisboa, 1649-028 Lisbon, Portugal

**Keywords:** resilience trajectory, psychological intimate-partner violence exposure, latent transition analysis, adolescents

## Abstract

Despite the serious emotional and social consequences of adolescents’ exposure to intimate-partner violence (IPV) and the high prevalence of this exposure, few analyses have focused on person-centered models or considered psychological IPV. Studies that address exposure to violence tend to focus on physical IPV. Therefore, in this study, we examine (across two waves) the trajectories of resilience among adolescents who have witnessed psychological IPV by conducting a latent transition analysis and predicting class membership through socio-demographic and individual-level protective factors. Using a sample of 879 (T1, fall 2020) and 770 (T2, spring 2022) adolescent Swiss students with mean ages of 11.74 (*SD* = 0.64) and 13.77 (*SD* = 0.53), we identified four distinct time-invariant resilience classes: comorbid-frustrated, internalizing-frustrated, comorbid-satisfied, and resilient. The classes characterized by some level of psychopathological symptoms and basic psychological-needs frustration were the most stable over time. Furthermore, we found the four typical resilience trajectories: recovery, chronic, delayed, and improving. Gender, socioeconomic background, and protective factors showed a significant prediction of class membership in wave 1, highlighting the importance of increasing sensitivity to psychological-IPV exposure on the one hand, and reinforcing the relevance of prevention in schools regarding the promotion of protective factors on the other.

## 1. Introduction

Promoting youths’ positive development is a central concern of a society and its educational system. Mental health problems are determinants of adverse outcomes across the lifetime, and these problems hamper the quality of life of children and youth [1], in addition to academic outcomes [2]. Several studies have shown a high prevalence of mental health problems in this life stage [3]; others indicate that these problems are often underrecognized and undertreated [4,5]. Identifying early signs of mental health problems is crucial to allocating resources and developing preventive interventions, namely through social and emotional learning and resilience promotion [6]. Schools, as one of the main contexts in the lives of children and adolescents, are spaces where significant manifestations of mental health difficulties frequently occur (e.g., social isolation and bullying). Because adolescence is an important and vulnerable period in development, and the prevalence of burdening experiences in the family and school systems continues to prevail, the need for targeted support for these adolescents at school and at home increases. 

Research has clearly indicated that a major risk factor for healthy development that youths face is exposure to family violence. Youths are at a significant risk of lifelong negative consequences that are also reflected in society [7,8], with intimate-partner violence (IPV) exposure being one of the most common familial burdens for them [9]. Violence resilience refers to a process whereby individuals develop adaptively despite experiences of violence that lead to the significantly increased likelihood of negative socio-emotional consequences [10]. According to Masten [11], resilient outcomes are characterized by competence at age-specific developmental tasks as well as an absence of psychopathological symptoms, despite risk exposure. Although resilience is seen as a process, in order to methodically capture a moment in time or a trajectory of resilience, resilience outcomes are measured. When defining resilience outcomes, we always assume that they are a snapshot of a process and not a final outcome. The present study draws on Deci and Ryan’s [12] self-determination theory, which provides extensive evidence that meeting basic psychological needs (autonomy, relatedness, competence) increases the ability and adaptability of individuals and reduces their vulnerability to psychopathology [13]. Psychopathological behaviors stemming from IPV exposure that have been particularly well studied include depressive [14] and aggressive behaviors [15], along with the lesser studied but equally important peer victimization [16]. 

In accordance with Ungar’s [17] three dimensions of resilience research, in this article we will analyze desired resilience outcomes (dimension 1) and promotive factors of these resilience outcomes at the individual level (dimension 2), as well as sociodemographic factors, despite witnessing psychological interparental violence as a form of risk exposure (dimension 3). From a resilience perspective, we focus on the socio-emotional development of youths in the school system who are exposed to psychological IPV in the family system, which is associated with an increased likelihood of negative outcomes and impaired socio-emotional development. As such, they are an ideal place to identify and address these difficulties [18]. 

### 1.1. IPV Prevalence and the Consequences of Exposure for Adolescents

In general, IPV is defined as stalking and physical, sexual, or psychological violence, the latter of which can take such forms as expressive aggression, coercive control, threats of violence, exploitation of vulnerabilities, and gaslighting [19]. Women and men are victims of IPV [20], in which perpetrators of violence can be (ex-) spouses or current or former partners [21], but women are more likely to suffer physically or sexually from IPV than men [9]. The prevalence rates of IPV decrease or increase depending on the measurement and use of criteria-based data [22]. Burrows et al. [9] found that physical IPV affects 40% of women and 15% of men worldwide. Looking specifically at psychological IPV, the prevalence rate for women is 41% and even as high as 51% in men. 

Although the scientific community agrees that children and adolescents exposed to IPV suffer far-reaching consequences, it seems difficult to find adequate prevalence data regarding specific youths’ exposure to IPV. Using a representative sample of 4000 individuals from the National Survey of Children’s Exposure to Violence, Finkelhor et al. [23] found that, overall, 5.8% had witnessed an assault between their parents, and 25% of youths ages 14–17 had been exposed to psychological IPV, which was the largest prevalence rate of any age group. A study on psychological-IPV exposure in emerging adulthood showed that 58% of participants had witnessed IPV in the previous year [21]. The overall lack of concrete prevalence rates specific to psychological IPV may stem from the fact that studies on domestic violence are neither numerous nor reflective of the severity of the suffering [20]. Exposure to psychological IPV is a more common sub-form of IPV in our society, with equally far-reaching consequences as exposure to physical IPV [24]. Analyses of psychological IPV have shown that such exposure can disrupt individuals’ emotional security, paving the way for the development of psychopathology [25,26].

The adverse consequences of IPV exposure for a variety of childhood and adolescent developmental outcomes have been well-documented in the literature [8,27,28,29]. In addition, the vulnerability of adolescents from high-conflict homes has been shown to predominate across a broad range of functional domains, such as the social, behavioral, emotional, and academic [30]. Systematic reviews and meta-analyses indicate that youths exposed to psychological IPV are more likely to exhibit externalizing symptomatology and the internalizing of behavior problems, in addition to posttraumatic stress symptoms [8,28,31], including flashbacks, the avoidance of traumatic experiences, resignation, and anxiety disorders [32]. Therefore, psychological-IPV exposure in adolescence is associated with an increased risk for developing behavioral, cognitive, social, and emotional difficulties; however, these research findings are highly dependent on the affected individuals’ age and developmental stage [28,33,34,35,36]. The impact of overt interparental conflict on youths has the largest effect when compared to covert conflict, withdrawal from conflict, and conflict frequency [37]. Moreover, the reports of youths are rarely used in the analysis of exposure to IPV, affecting not only the prevalence rates but also the analysis results. Consequently, according to Hungerford et al. [29], including youths’ reports of their exposure to IPV suggests a way to increase the understanding of youths’ adjustment to IPV exposure. Having highlighted the consequences of psychological-IPV exposure, the next section defines what resilience means in the context of psychological-IPV exposure.

### 1.2. Defining Resilience despite IPV Exposure

According to Pietrzak & Southwick [38], when defining resilience, it is essential to specify whether it is considered a trait, a process, or an outcome. The understanding of resilience as a trait is strongly criticized in current research because the entire responsibility for healthy development is attributed to the individual, which can quickly lead to victim blaming [39,40]. The consensus clearly leans toward resilience being a process and outcome that can change considerably over time [41,42] as new vulnerabilities and strengths emerge with varying circumstances [43,44]. Therefore, we followed Masten [11], who defines resilience as the result of dynamic processes that change over time, enabling a dynamic system to adapt to adversity that threatens its function and development. Masten uses the term system with the intent that the definition of resilience can be applied not only to the individual but also to other systems, such as the family, school, or community. Accordingly, it is crucial to examine resilience system and domain specifically, as well as developmental stage dependently, because it is a very broad concept. As Bonanno et al. [45] note, resilience has a temporal moment, so it is necessary to specify what type of adversity is involved when resilience is operationalized. Bonanno et al. [45] distinguish between emergent resilience and minimal-impact resilience. Minimal-impact resilience describes resilient processes that can arise after acute exposure to an adversity. Emergent resilience, on the other hand, describes trajectories that emerge as a result of chronic adversity, such as child abuse [46]. We follow the assumption of Pietrzak & Southwick [38], who put forward the understanding that resilience occurs on a continuum rather than being dichotomous (i.e., resilient and non-resilient) and therefore can vary across life domains. Therefore, we will examine emergent IPV-exposure-resilience trajectory outcomes because developmental trajectory is regarded as one of the most relevant research topics in the social and behavioral sciences [47]. 

### 1.3. Domains for Non-Normative Development in the Face of Psychological IPV 

Looking at domain-, system-, and developmental-specific resilience regarding psychological-IPV exposure consequences, several outcomes are related to psychological-IPV exposure and are therefore crucial in operationalizing specific resilience outcomes. Psychological-IPV exposure is seen to correlate strongly with youths’ anxious and aggressive behavior problems, which seem to increase alongside youths’ self-blame for inter-parental conflict [48]. Aggression against peers because of psychological-IPV exposure through social-learning theory is further understood as a social behavior that is learned from parents and takes place through observation [49]. Regarding peer victimization, social-learning theory postulates that through observation of aggression in interparental conflicts, youths may come to accept bullying as a legitimate form of social interaction [50]. 

In terms of age-specific developmental tasks, basic psychological-need theory as a self-determination (SDT) mini-theory is suitable for assessing adolescents’ development because a central aim of SDT is to explain the promoting conditions of optimal development of the self [51]. As Vansteenkiste et al. [52] note, numerous meta-analyses have confirmed Deci and Ryan’s [12] basic psychological-needs theory, which describes autonomy, competence, and relatedness as three fundamental psychological needs, to be a universal, critical resource for an individual to self-organize, adapt, and thrive. There is extensive evidence that meeting those needs increases individuals’ abilities and adaptability, reduces vulnerability to psychopathology, and is an aspect of resilience that facilitates growth during challenges [13,52]. The frustration of these needs, on the other hand, can contribute to psychopathology [53]. The need for autonomy describes the need for agency and authenticity characterized by the skills of self-regulation and volition. Especially in adolescence, autonomy support seems to play a crucial role because it is a critical development period in which independence gains importance but the need for support and nurturance remains [51]. Experiences of interest and value support autonomy, and external control undermines it [54]. The need for relatedness is strongly connected to autonomy and describes “the strength and quality of one’s connection to others in a given context” ([51], p. 817). As Ryan et al. [51] put it, the quality of relatedness suffers with the absence of autonomy because a lack of acceptance of one’s real self leads to superficial and contingent connections. The need for competence refers to the need to feel effective in one’s interaction with the environment [12]. A well-structured environment offering optimal challenges, positive feedback, and chances for further development can foster the feeling of competence [51]. 

### 1.4. Resilience Trajectories of Youth’s IPV Exposure

If we look at the existing empirical research on resilience patterns, we find that similar patterns are found again and again. Cameranesi [55], who specifically examined IPV exposure from a psychopathology perspective, found four adjustment profiles: resilient, multiple severe problems, multiple mild problems, and externalizing problems. Kassis et al. [10] also found four resilience and well-being patterns despite physical parental violence experiences, which they based on the dual-factor model of mental health, which include the resilient, troubled, vulnerable, and non-resilient. Janousch et al. [56], who examined resilience trajectories in Switzerland without accounting for risk factors, found three trajectories: resilient, non-resilient, and untroubled. 

Considering the changes over time, i.e., the transition from one category at wave 1 to another category at wave 2, resilience trajectories across adversities fall into four categories. Bonanno [57,58] observed the prototypical patterns of normal functioning disruption that occur over time as a result of personal loss or potentially traumatic events, and found that resilience pathways do not necessarily follow the same course but can be divided into four categories. The resilience trajectory includes individuals who continue to maintain relatively healthy levels of psychological and physical functioning despite potentially traumatic experiences. Distinct from this trajectory, according to Bonanno [57], is the recovery trajectory, in which normal functioning temporarily diminishes into underlying psychopathology, such as depressive symptoms, and then gradually returns to healthy levels after several months. In the chronic trajectory, psychopathological symptoms persist over time, and individuals in the delayed trajectory experience a deterioration in adaptive development over time. Researchers have demonstrated these four trajectories following potential trauma to varying degrees in many studies [59]. In their review study, Galatzer-Levy et al. [59] showed a substantial overall structural and numerical consistency across studies. The most common trajectories observed were resilience, recovery, and chronic stress. The delayed trajectory was the least frequently observed of these four trajectories, reportedly reflecting its rarity in the population. Masten and Narayan [60] found three trajectories, which they called pathways following acute trauma, namely stress resistance, breakdown and recovery, and posttraumatic growth. Oshri et al. [61] found the following four social resilience trajectories after child maltreatment: stress-resistant, emergent-resilient, breakdown, and unresponsive-maladaptive. Meijer et al. [62], who explicitly studied youths with IPV exposure, reported the resilient, moderately stable, struggling, improving, and highly adaptive trajectories, with only the struggling trajectory reaching a dysfunctional level. 

### 1.5. Protective and Sociodemographic Factors as Predictors

It was empirically established decades ago that only a small proportion of children and adolescents who have experienced violence can be considered to be on the particularly unstable resilience trajectory [63,64]. As Bonanno et al. [45] summarize, the factors that play a role in adolescents becoming and remaining resilient can be found at the individual, family, and community levels. Measuring resilience at the family and community levels is currently a very difficult endeavor, with little research on the subject and cultural and socioeconomic differences playing a role. Particularly significant factors at the individual level that can promote resilience are high self-esteem and a positive self-image, because a strong, optimistic self-efficacy strengthens one’s ability to adapt to the challenges of adverse circumstances [45]. Results indicate that perceived self-efficacy and support from others tend to be related to psychological adjustment [65]. In terms of individual characteristics, analyses of IPV exposure and resilience outcomes have yielded evidence that maternal education level predicted resilience profiles, although gender and ethnicity did not always show significant results [66]. Kassis et al. [10] found that only female gender predicted one’s assignment to the resilient group versus the non-resilient group in two waves, and migration background (In Switzerland, due to migration policy and the cultural context [67], people with a migration background are either foreign nationals or naturalized citizens according to the Federal Statistical Office [68]. Individuals born in Switzerland whose parents were born in Switzerland and individuals with birth citizenship whose parents were born abroad are excluded.) and socioeconomic background did not.

### 1.6. Present Study

We intended to analyze the resilience trajectories of youths exposed to psychological IPV for at least 12 months by applying a longitudinal person-centered model because it can provide new insights into specifically targeted prevention and intervention programs in middle schools. We intended to examine the little-noticed resilience trajectories of psychological IPV, which are associated with consequential effects as severe as other forms of violence exposure [30]. Consistent with Ungar’s [17] three dimensions of resilience research, we examined specific risk exposure, considered specific outcomes focused on the consequences of psychological-IPV exposure, and analyzed promoting factors. Our overall goals were to demonstrate the great impact of the exposure to violence and to identify protective factors that can be promoted at the school and family levels. Based on existing theories of resilience in general and specific research findings on violence resilience, we aimed to answer the following questions and assumptions: (1)What are the trajectories of adolescents’ non-dichotomous resilience outcomes across the two waves despite psychological-IPV exposure? Following insights from past studies [45], we expect to identify at least four resilience trajectories over time: a resilient, a chronic, a delayed, and a recovery trajectory.(2)How stable are IPV exposure classes over two waves? We expect the resilient class to be the most unstable class [63,64]. (3)Do pattern affiliations differ by protective factors and sociodemographic characteristics? We expect the trajectory affiliations to differ for gender [10] and socioeconomic background [66], as well as self-efficacy, self-esteem, and social competence [45].

## 2. Materials and Methods

### 2.1. Participants

The analyzed data come from a longitudinal sample of a broader study of adolescents’ violence resilience conducted in the fall of 2020 (the beginning of grade 7; wave 1) and the spring of 2022 (the end of grade 8; wave 2). The sub-samples, consisting of adolescents who reported having experienced psychological IPV in the past 12 months, of both waves of the representative convenience sample consisted of 879 (wave 1: 58.3% of the overall sample) and 770 (wave 2: 58.9.% of the overall sample) grade 7 students from Switzerland. There are currently no statistics available on psychological IPV experiences in Switzerland, making it difficult to determine the representativeness of reported experiences. This gap in data collection highlights the need for further research and advocacy to shed light on this important issue. However, with the recent adoption of the 2022 motion, it is hopeful that there will be efforts towards the collection of statistical data on psychological IPV experiences in the future [69]. Female (wave 1 sex = 52.6%, wave 2 sex = 54.3%) and male participants (wave 1 sex: 47.4%, wave 2 sex: 45.7%) anonymously completed an approximately 60-min online questionnaire that research team members introduced verbally on the day of the study in the respective classrooms. As the gender distribution in Switzerland is relatively balanced, with 50.4% women, the IPV subsamples contain slightly more women than would be expected in the population [70]. Of the participating students, 51.6% (wave 1) and 51.8% (wave 2) were not Swiss citizens. In Switzerland, the Federal Statistical Office only covers the permanent resident population aged 15 and over; however, it is worth noting that 39% of the permanent resident population has a migration background, and 52% of children aged 7–14 live in households where at least one parent has a migration background [68]. The participants’ mean age was 11.7 in the first wave (*SD* = 0.64), and 13.8 (*SD* = 0.43) in the second. In the first wave of data collection, 37.5% of participants had a low SES, 46.7% had a medium SES, and 15.6% had a high SES. In the second wave, 32.8% had a low SES, 48.8% had a medium SES, and 15.4% had a high SES. While a direct comparison with the population was not available from the Federal Statistical Office, in 2021, 12.6% of the Swiss population had a low educational level, 42.4% had a medium educational level, and 45% had a high educational level [71]. Although education level is not equivalent to SES, we can assume that our sample of adolescents with psychological IPV experiences had a lower than average SES, confirming existing studies [72,73].

The research ethics committee of the University of Teacher Education FHNW approved the project. Once the ethics committee approved the study, we reached out to the cantons of north-western Switzerland, which is a German-speaking area, as well as all secondary school administrations in all four cantons. We then contacted all class teachers, who then obtained consent forms from parents and students. Participation was voluntary and contingent upon signed declarations of consent and without incentives. This was a non-exhaustive sample, meaning that only students who chose to participate were included in the sample.

### 2.2. Measures

#### 2.2.1. Prevalence of Exposure to Psychological IPV

The three-item indicator adapted from Mayer et al. [74] (loud shouting, verbally insulting, offending) of the last-year prevalence of IPV exposure indicates that adolescents reported having been exposed to psychological IPV at least once in the 12 months before the fall of 2020. A sample item is *I witnessed how my parents shouted at each other very loudly.* We categorized responses as “no; never” (1); and “yes, at least seldom in the last 12 months” (2). Cronbach’s alpha was 0.86 for wave 1 and it was 0.91 for wave 2.

#### 2.2.2. Symptoms of Depression and Anxiety

To assess symptoms of depression and anxiety, we used 24 items from the Hopkins Symptom Checklist [75]. We dropped one item, loss of sexual interest or pleasure, from the original 25-item scale due to the participants’ young age. The participants rated the items on a 4-point Likert scale ranging from (1) “not at all” to (4) “extremely”, with higher scores indicating more severe anxiety and depression symptoms. We calculated a median split (MED_wave 1 = 1.63, MED_wave 2 = 1.77) to categorize scores as low (1) and high (2) for the LTA. The Cronbach’s alpha was 0.96 for wave 1 and 0.97 for wave 2.

#### 2.2.3. Frequency of Overt Aggression and Overt Victimization toward and from Peers

To assess overt aggression and victimization, we used the Overt Aggression subscale of the German Self-Report Behavior Aggression-Opposition Scale, which consists of the following five items [76]: teasing to make angry, physically pushing around, threatening to hurt someone physically, name calling/insulting, and physically hurting. Respondents rated the items on two 4-point Likert scales according to whether they were perpetrators or victims of overt aggression: (1) never happened, (2) once or twice a month, (3) once a week, and (4) more than once a week since school started, with higher scores indicating more frequent perpetration or victimization. For LTA, we calculated a median split (MED_wave 1_perpetrator = 1.2, MED_wave 2_perpetrator = 1.4, MED_wave 1_victim = 1.2, MED_wave 2_victim = 1.4) to categorize scores as low (1) and high (2). Cronbach’s alpha was 0.80 for overt aggression in wave 1 and 0.86 for overt aggression in wave 2. Cronbach’s alpha was 0.84 for overt victimization in wave 1 and 0.88 for overt victimization in wave 2.

#### 2.2.4. Basic Psychological Needs Satisfaction and Frustration Scale

The Basic Psychological Needs Satisfaction and Frustration Scale, a three-subscale measure of autonomy, competence, and relatedness, is a short scale with three items, each following Kassis et al. [10]. The three subscales included a four-point Likert scale, from strongly disagree (1) to strongly agree (4). For the LTA, we performed a median split for each subscale and categorized the scores as (1) lower, and (2) higher levels of autonomy (MED_Wave 1 = 3.0, MED_Wave 2 = 3.0), competence (MED_Wave 1 = 3.0, MED_Wave 2 = 3.0), and connectedness (MED_Wave 1 = 3.33, MED_Wave 2 = 3.0). Cronbach’s alpha for connection was 0.81 for wave 1 and 0.86 for wave 2; for competence, it was 0.79 for wave 1 and 0.86 for wave 2; and for autonomy, it was 0.72 for wave one and 0.80 for wave 2.

#### 2.2.5. Predictors of Class Memberships

We categorized adolescents by gender (1 = female; 2 = male) and migration background (1 = Swiss citizen; 2 = first- or second-generation immigrant without Swiss citizenship).

We calculated socioeconomic status using information about the *highest level of school education for both parents* (1 = primary school/junior high school, 2 = vocational education/general high school certificate, 3 = university degree/higher education), *financial background* using the item educational and computer-related belongings [77], and *number of books in household* and *number of own books* [78]. We developed a composite score from the three scales and categorized them as low = 1, medium = 2, and high = 3.

We measured self-efficacy using the General Self-Efficacy Scale from Jerusalem and Schwarzer [79]. The participants rated the six items on the short scale (Cα_wave 1 = 0.88, Cα_wave 2 = 0.92) on a four-point Likert scale from (1) not true to (4) completely true. For the LCA/LTA, we performed a median split (MED_wave 1 = 2.83, MED_wave 2 = 2.83) to categorize the variable as a (1) lower or (2) higher level of self-efficacy

We assessed self-esteem using the five-item Rosenberg Self-Esteem Scale [80] short scale, with higher scores indicating higher self-esteem. The participants rated the items on a 4-point Likert scale ranging from (1) not at all to (4) extremely (Cα-wave 1 = 0.90, Cα-wave 2 = 0.92). For the LCA/LTA, we performed a median split (MED_wave 1 = 3.00, MED_wave 2 = 3.00) to categorize the answers as (1) lower and (2) higher self-esteem.

To measure social competence, we used the Brief Social Competence Scale from Anderson-Butcher et al. [81], which measures perceived social competence with four items. Adolescents rated the items on a four-point Likert scale ranging from (1) strongly disagree to (4) strongly agree. Cronbach’s alpha was 0.82 for Wave 1 and 0.86 for Wave 2. For the LCA/LTA, we performed a median split (MED_wave 1 = 3.25, MED_wave 2 = 3.00) to categorize it as (1) lower or (2) higher social competence.

### 2.3. Analytical Plan

Our goal was to create a longitudinal finite mixture model (FMM) because it offers great flexibility in modeling complex data. Examples of longitudinal FMMs include growth mixture modeling, latent class growth analysis, and latent transition analysis (LTA). The advantage of LTA is that, in addition to modeling trajectories, discrete time-invariant latent states allow for the examination of transition probabilities from one state to another at various waves [82]. To answer the research question, we first conducted descriptive analyses of the variables at hand, in addition to *t*-tests to check for any mean differences among the indicators. The attrition of the IPV samples from wave 1 (*n* = 879) to wave 2 (*n* = 770) was 12.4%. Little’s chi squared statistic showed that the missings were completely random (χ^2^ = 230.595, df = 249, *p* = 0.79). Due to the missings caused by COVID-19 and the nonnormal data, we performed all calculations with SPSS Version 24 and Mplus Version 8.4, using the listwise deletion method and maximum-likelihood estimation with robust standard errors. Subsequently, we calculated the latent class analysis separately for the two waves and used model fit indices to select the model with the best number of classes [83]. The Akaike information criterion (AIC), the Bayesian information criterion (BIC), the Sample-Adjusted BIC (aBIC), entropy, posterior classification probabilities, (adjusted) the Lo, Mendell, and Rubin Test (LMR), and the bootstrapped likelihood ratio test (BLRT) are the most used selection criteria. Simulation studies showed that adjusted LMR, BLRT, and BIC are the most reliable selection criteria for the number of profiles regardless of sample size, where aBIC performs as well as BLRT except with small samples [84]. Since the four-class solution showed the best fit in both models, we calculated the latent transition analysis with four classes. The next step was to test whether the two waves were time-invariant; that is, whether the classes exhibited measurement invariance over time [85]. Time invariance is not required for the calculation of LTA because classes can change over time. Nevertheless, when determining the model for the LTA, it is important to test for measurement invariance over time to capture the correct transition probabilities. When we compared the full-time invariant model and the full-time variant model using a chi squared test (∆χ^2^ = 41.89, df = 24, 0.01 < *p* < 0.05), we found that the two models differed significantly (*p* > 0.05), but not highly significantly (*p* < 0.01). When we compared the models using information criteria, lower BIC and aBIC values indicated that the full-time invariant model represented the data better. Following McElroy et al. [86], we assumed full measurement invariance, and we computed the final LTA with the time-invariant model. In the final step, we predicted the probability of class membership for wave 1 based on the predictor variables using the three-step method of Mplus.

## 3. Results

### 3.1. Descriptives and t-Tests

The mean age of the adolescents was 11.7 (*SD* = 0.64) years in wave 1 (fall 2020), and 13.8 (*SD* = 0.43) years in wave 2 (spring 2022). Table 1 shows the mean comparison of the resilience indicators. For the psychopathology indicators of depression/anxiety, peer aggression, and peer victimization, there was a slight decrease in mean scores between waves 1 and 2, but there were no significant mean differences between the two waves. The picture was different for the psychological-needs-satisfaction indicators. The mean values of autonomy and relatedness significantly decreased from wave 1 to wave 2, and, interestingly, the mean value of competence significantly increased. However, all identified mean differences have a negligible effect size, as Cohen’s d shows (see Table 1). For the predictors, all but gender show significant mean differences, but the effect is again very small. The largest effect has a decreasing mean in social competence. Self-esteem increased significantly from wave 1 to wave 2, but social competence and self-efficacy decreased.

### 3.2. Separate Latent Class Analysis for Both Waves

For both waves, the four-class solution has proven to be the most optimal (see Table 2). In the first wave, the AIC, BIC, and aBIC values were lowest for the four-class solution and the VLMR, LMR, and BLRT tests showed that the four-class solution fitted better than the five-class solution. In the second wave, the AIC score was lower for the five-class solution, but the BIC and aBIC values, which are considered the most reliable criteria, [84] favored a four-class solution, as did the VLMR, LMR, and BLRT tests. Therefore, we chose the four-class solution for both waves.

The four-class solution for both waves resulted in the classes shown in Figure 1 and Figure 2. Due to visually similar patterns, we gave the classes in both waves the same names. Figure 1 shows the resilience classes for wave 1. A class we called *resilient*, which accounted for 18.3% of the sample, had low levels on the psychopathological indicators and high levels of relatedness, competence, and autonomy. In contrast, 38.4% had high levels on all psychopathology indicators and low levels of psychological-needs satisfaction; we labeled this class *comorbid-frustrated*. One class had high levels on all indicators, indicating high psychopathology symptomatology, but also high levels of psychological-needs satisfaction, a class we termed *comorbid-satisfied* (23.5%). The final class, comprising 19.8% of the wave 1 sample, had low psychological-needs satisfaction, with the lowest competence levels in all four classes and low scores on externalizing behavior, but high levels of depression/anxiety, so we named it *internalizing-frustrated*. In the second wave, the classes behaved similarly (see Figure 2). A total of 21.8% of the adolescents were in the resilient class, 29.2% in the comorbid-frustrated class, 14.2% in the comorbid-satisfied class, and 34.8% in the internalizing-frustrated class.

As shown in Section 2.3, the two models differed significantly, but not highly significantly (∆χ^2^ = 41.89, df = 24, 0.01 < *p* < 0.05). Following McElroy et al. [86], we assumed and used full measurement invariance for the LTA model because lower BIC and aBIC values indicated that the fully invariant model better represented the data (see Table 3).

The latent transition probabilities of violence resilience classes (see Table 4) made clear that the most stable class, with a probability of 64.8% of the respondents to remain over two school years, was the internalizing-frustrated class. Following this was the comorbid-frustrated class, at 53.4%; the resilient class, at 50.8%; and the comorbid-satisfied class (the least stable class), at 32.8%. Overall, the probability of youth developing resilience during the two school years was 45.7%, and the probability of not maintaining resilience was 49.2%. The smallest probability of becoming resilient in wave 2 was for youths who were comorbid-frustrated in wave 1 (6.8%). The probability of becoming resilient in wave 2 was 17% for youths who showed high levels of symptoms and high levels of basic-needs satisfaction, and 21.9% for youths who showed high levels of internalizing symptoms only, but low levels of basic-needs satisfaction. The transition probability was highest for youths in the comorbid-satisfied class in wave 1 to move to the comorbid-frustrated class in wave 2 (39.1%), followed by youths in the resilient class in wave 1 moving to the internalizing-frustrated class in wave 2 (34.1%), and youths in the comorbid-frustrated class in wave 1 moving to the internalizing-frustrated class in wave 2 (27.1%). The lowest transition probability was 1.9% for youths moving from internalizing-frustrated in wave 1 to comorbid-satisfied in wave 2, followed by the transition probability of 4.3% of moving from wave 1 (resilient) to wave 2 (comorbid-frustrated).

As Figure 3 shows, 11% of the total 21% of youths who were in the resilient class in wave 1 were also in the resilient class in wave 2. The remaining 10% of the overall sample who were in the resilient class in wave 1 transitioned to a non-resilient class in wave 2. Of the total 17.1% of the overall sample who were in the comorbid-satisfied class in wave 1, only 5.7% of youths remained in that class in wave 2. A total of 2.8% moved up to the resilient class in wave 2, and the remaining 8.6% had worsened development. Of the total 28.4% of youths in the internalizing-frustrated class, 19% remained in the same class over the two school years, and 6% developed resilience. The remaining 3.4% developed externalizing symptoms in addition to internalizing symptoms. Of the total 33% of adolescents who were in the comorbid-frustrated class in wave 1, 20% remained in the same class in wave 2, and 2% developed resiliently during the two school years. The remaining 11% developed more positively either in terms of psychopathology or need satisfaction, but not both.

In the recovery trajectory, which is characterized by initial symptoms followed by recovery [57], about 11% of adolescents developed from having initial symptoms in wave 1 to being resilient to psychological IPV exposure in wave 2. These are the percentages in Figure 3 that move from one of the non-resilient classes in wave 1 to the resilient class in wave 2; that is, the 15, 40, and 19 individuals who moved from comorbid-frustrated, internalizing-frustrated, and comorbid-satisfied in wave 1 to resilient in wave 2, respectively. About 11% of youths were in a *resilient trajectory*. These individuals did not develop psychopathological symptoms during these two school years, and their basic psychological needs were satisfied; that is, the adolescents who were in the resilience class in waves 1 and 2 [57]. An additional 10% of youths were initially resilient in wave 1 but developed symptoms and frustration with regard to their basic psychological needs in wave 2 (except in the case of the comorbid-satisfied class in wave 2), and were on the *delayed trajectory*. However, some youths developed positively in one way or another, even if their development was not enough for resilient development. Overall, 11% of adolescents showed positive development in terms of either their psychopathological symptoms (from comorbid-frustrated to internalizing-frustrated), or their basic need satisfaction (from comorbid-frustrated to comorbid-satisfied), and were therefore on a *slightly improving trajectory* [62]. However, the remaining adolescents, who accounted for more than half of the sample (57%), either showed chronically psychopathological symptoms or their condition worsened over two school years; they were therefore on a *chronic trajectory* [57].

To determine which protective factors and individual characteristics increase the likelihood of adolescents belonging to a resilient class, we conducted a multinomial logistic regression analysis for wave 1, following Lanza et al. [83]. Female gender increased the chances of being in the resilient class compared to the comorbid-satisfied and comorbid-frustrated class, but not the internalizing-frustrated class (see Table 5). Male adolescents were more likely to be in the comorbid-satisfied and comorbid-frustrated classes than in the internalizing-frustrated class. With higher socioeconomic status, adolescents were more likely to be in the resilient, comorbid-satisfied, or internalizing-frustrated class than in the comorbid-frustrated class. Higher self-esteem increased the chances of being in the resilient class compared to the other three classes, and of being in the comorbid-satisfied or internalizing-frustrated class than in the comorbid-frustrated class. Higher self-efficacy increased the chances of being in the resilient class compared to the internalizing-frustrated or comorbid-frustrated class, but not the comorbid-satisfied class. Interestingly, with increasing self-efficacy, the chances of being in the comorbid-satisfied class increased compared to the internalizing-frustrated or comorbid-frustrated class. The same goes for social competence. Migration background did not predict class membership.

## 4. Discussion

In the present study, we went beyond a dichotomous understanding of resilience by taking a person-centered approach and uncovering resilience trajectories considering psychopathological symptoms and adolescents’ three basic psychological needs with a multidimensional understanding of resilience [17]. We examined the risk-specific trajectories of resilience in adolescents despite psychological-IPV exposure and the prediction of protective factors, as well as individual characteristics of youths’ membership in these classes. We asked three research questions that we will now discuss in detail.

Overall, our results showed high last-year prevalence rates (about 60% for both waves) of psychological-IPV exposure among adolescents, which is consistent with the findings of Black et al. [21], but at the same time were relatively high, possibly because researchers have conducted few studies on the prevalence and incidence of psychological-IPV exposure [20]. We specifically interviewed adolescents and not their parents, possibly leading to under-reporting in the case of parent reports [87], and the rate may have increased during COVID-19 [88]. Regardless of the reasons for the high prevalence, our study highlights the urgent need for representative prevalence and incidence measurements of IPV exposure. The high prevalence and the lack of empirical evidence on psychological IPV clearly indicate that family violence does not receive enough attention, as Sigurdsson [20] points out. Regarding health care institutions, Sigurdsson [20] states that there is a need to remove barriers to reporting violence, raising awareness of characteristics that indicate possible exposure to IPV, and promote education about IPV. As adolescents spend most of their time at school, we believe that raising awareness of possible exposure to violence can help to sensitize school staff and demonstrate the importance of reporting violence.

To address and improve prevalence rates of family violence, it is necessary for various stakeholders to be involved. Caregivers, educators, and professionals should understand the negative impact of stigma and shame on children and adolescents who have experienced parental violence, as this affects their reporting [89]. Furthermore, governments have a crucial role to play in preventing and responding to family violence through policies, legislation, and funding. They can develop and implement programs and services that support victims, raise awareness, and hold perpetrators accountable through the explicit prohibition of family violence, which is still lacking in Switzerland [69].

Our first research question concerned the trajectories of adolescents’ non-dichotomous resilience outcomes across two waves, despite psychological-IPV exposure. Using LCAs and LTAs, we identified four latent classes from wave 1 (autumn 2020) and wave 2 (spring 2022) following IPV exposure, encompassing the absence of psychopathology and age-specific developmental tasks [11]. We examined these two conditions for adaptive functioning using depression/anxiety, peer aggression, and peer victimization, as well as the three indicators of autonomy, relatedness, and competence. We labeled the four time-invariant classes comorbid-frustrated, internalizing-frustrated, comorbid-satisfied, and resilient according to psychopathological [8,28,31] and self-determination theory [53] considerations. Because about one-fifth of the sample was in the resilience class in both waves, and the largest proportion of youth were in the class with the most negative developmental outcomes, our results are consistent with previous findings [10,90]. Differences in prevalence rates have to do with differences in the composition of the measurement instruments, the operationalization of resilience, and the varying samples [90,91]. We assumed that we would find at least four resilience trajectories over time: the resilient, chronic, delayed, and recovery [57]. We confirmed these findings, and in addition to the listed trajectories according to Bonanno [57], we located a *slightly improving* trajectory according to Meijer et al. [62], in which adolescents, however, remained non-resilient despite positive development. We consider this trajectory particularly important in terms of future research regarding protective factors because it involves approximately 11% of the adolescents in our sample who are not resilient but who nonetheless developed positively in terms of either psychopathology or basic need satisfaction during the two years of secondary school. However, our results also showed that more than half of the adolescents exposed to psychological IPV were on a chronic trajectory, i.e., either they were consistently non-adaptive, or their situation had worsened in the last two school years. In their review, Galatzer-Levy et al. [59] explain the high prevalence rate for the chronic trajectory using a longitudinal design, and consider the fact that the risk factor (in our case, IPV exposure) is chronic and not acute. It is essential to emphasize that the last two years were quite unique due to the COVID-19 pandemic, as several restrictive measures and lockdowns were imposed on the population, with significant consequences for mental health. Researchers have thoroughly documented the pandemic’s impact on adolescents in several systematic reviews and meta-analyses [92,93,94,95,96,97]. A high prevalence of mental health problems, namely anxiety and depression, experienced during this period was subsequently verified. The cumulative effect of pandemic-related stressors was associated with increases in internalizing and externalizing symptoms [98]. The association of pandemic-related stressors with other significant stressors, such as IPV exposure, could also play a role in the high prevalence rate for the chronic trajectory. For instance, in the cumulative risk scope, Gutman et al. [99] mention that in addition to aspects such as the number, degree, and severity of risks, the persistence of risk factors during childhood and adolescence is an important determinant of developmental trajectories.

Our second research question addressed the stability of these IPV-exposure resilience outcomes over two school years, and we assumed that the resilient class would be the most unstable one [63,64]. We could not confirm this assumption, however, because the most unstable class with a probability to remain in that class in the second wave (of about 33%) was the comorbid-satisfied class. The resilient class was the second most unstable class, in which there existed only about a 51% probability to be remain resilient for two school years. The most stable class, with a probability of almost 65% to remain in the class, was the class with elevated levels of internalization and low psychological need satisfaction: the internalizing-frustrated class. Youths in the comorbid-frustrated class in wave 1 had the lowest likelihood of becoming resilient in wave 2, and youths in the internalizing-frustrated class in wave 1 had the highest likelihood. The likelihood of transition in wave 1 to move to the comorbid-frustrated class in wave 2 was highest for youths in the comorbid-satisfied class. Interestingly, the probability of decreasing depressive and anxious symptoms and increasing basic needs satisfaction was higher over the two school years than for that of decreasing internalizing and externalizing symptoms.

Our third research question concerned trajectory affiliation based on protective factors and sociodemographic characteristics. We assumed that trajectory affiliations would differ by gender [10] and socioeconomic background [66], self-efficacy, self-esteem, and social competence [45]. Therefore, as a last step, we conducted a multinomial logistic regression with socio-demographic variables and the protective factors self-esteem, self-efficacy, and social competence. The protective factor for being in a resilient class in wave 1 was female gender, but only compared to the comorbid-satisfied and comorbid-frustrated classes, not for the internalizing-frustrated class. Male adolescents had higher odds of being in the resilient class than in the internalizing-frustrated class. These findings are consistent with existing results [10]. Higher socioeconomic status further increased the odds of being resilient compared to those of being in the comorbid-frustrated class, confirming Cameranesi’s [66] findings. Migration background had no predictive significance.

Regarding individual modifiable protective factors, self-esteem increased the likelihood of being in the resilient class across all classes. For self-efficacy and social competence, this was only the case compared to the internalizing-frustrated or comorbid-frustrated class, not the comorbid-satisfied class, possibly because self-efficacy and social competence are strongly associated with the satisfaction of basic psychological needs, but confirming this would require more detailed analysis in future studies. Following Ryan et al. [51], the self-determination mini-theory of basic psychological needs has proven useful in operationalizing age-specific developmental tasks in the framework of resilience theory. To protect adolescents from the consequences of IPV exposure, the present article addresses several issues: First, the prevalence of IPV exposure should be researched more thoroughly, and society should be sensitized to the strong consequences of IPV exposure. As a further step, prevention efforts in schools should focus on promoting self-esteem, self-efficacy, and social skills, because they significantly increase the chances of being in the resilient class, and they can also be promoted in the school system [100].

Providing early mental health identification programs in school settings, such as universal screening and comprehensive clinical assessments, can improve the mental health care for children and adolescents in need, in addition to their academic performance [101]. In this scope, teacher training and support designed to identify and adequately refer children and adolescents [102], and parent training concerning awareness, early identification, and intervention needs [6], are fundamental. The mental health literacy programs for the various actors involved can also play an important role in increasing mental health knowledge and promoting positive attitudes toward students’ mental illness [103], reduce stigma-related attitudes and behaviors, and increase teachers’ confidence in helping students [104,105]. As early and enduring exposure to IPV can cause harm to the brain development and behavior of children [106,107], the negative effects may not be immediately evident, leading to delayed effects that can emerge from months to years after the exposure [108]. Given this knowledge, there is a strong argument for early intervention to mitigate the immediate impacts of IPV exposure and prevent future problems [106].

However, there are still many open questions about the relationship between violence exposure and its impact on adolescence, particularly regarding the appropriate time and method to intervene in order to promote well-being [109]. Some studies claim that there is insufficient evidence about the positive impact of interventions due to methodological issues [110,111], while others suggest that interventions such as parent-focused IPV programs or interventions with mothers and children, either jointly or separately, can mitigate or even nullify the negative impacts of IPV exposure [90,107,109,112]. It is important to note that universal interventions may not be sufficient for adolescents who have experienced certain forms of violence, such as psychological IPV [113]. Previous interventions for women who have experienced IPV are often non-specific and not theoretically based, which can lead to poor outcomes. Children and adolescents are even less likely to be included in these interventions, highlighting the need for more research and tailored interventions for this population [114]. To address the underlying factors of IPV, such as gender inequality or socioeconomic deprivation, some authors argue for the promotion of policies and the coordination of targeted services within the individual’s context (e.g., home, school, community) [115]. For example, at the school level, interventions tailored to the adolescent’s needs that include both trauma-specific and non-trauma-specific content, as well as the creation of a positive school climate and safer school environments, are recommended [107]. Some reviews suggest the importance of early intervention through universal programs and the use of an adaptation-based approach to resilience in tandem with a strengths-based approach to promote positive development for those children and youth growing up in adversity [109,116,117].

As a global burden, family violence is a significant public health problem that has a destabilizing effect on the normative development of future generations, and thus fosters further violence, inequality and misery. Future research should therefore focus on developing a more comprehensive understanding of the various factors that contribute to the resilience of young people, their families, and society, by examining the extent to which young people experience violence and its consequences in depth. Even if schools are an appropriate place to educate young people about violent behavior and experiences, the first step is to train teachers so that they have the positive attitudes, materials and skills to bring about these changes. In addition, there is a need to explore the concept of family resilience, which has received relatively little attention in resilience research, and which will be instrumental in reducing the prevalence of family violence, mitigating its negative effects and improving the resilience of victims and their families.

## 5. Limitations

Although we showed that psychological-IPV exposure is more common than currently shown empirically, and that resilience trajectories following trauma can be applied to IPV exposure and serve to identify important protective factors that can be promoted, there are some limitations to this study.

Because psychological IPV is a chronic condition, it is unclear as to how long youths have been exposed to it. As Bonanno et al. [45] note, measuring baseline levels of psychological adjustment is notoriously difficult because pre-event assessments are often unavailable. Therefore, inclusion of the baseline IPV exposure would be of interest for future research.

The present sample consists of adolescents who have experienced at least 12 months of psychological-IPV exposure. The possibility that additional poly-victimization has occurred cannot be excluded, and in fact this is likely the case. It would therefore be necessary to consider the same results from the perspective of poly-victimization. Nevertheless, the present results are a first step in an important direction, especially because studies show that IPV exposure brings its own consequences [118].

For measurement invariance reasons, it is difficult to make group comparisons with participants who have not experienced IPV exposure; this is a known problem that results in control groups rarely being included (see [67]). Therefore, latent resilience is always determined in relation to the sample, which means that the present study should be replicated in various cultural contexts.

## Figures and Tables

**Figure 1 ijerph-20-05676-f001:**
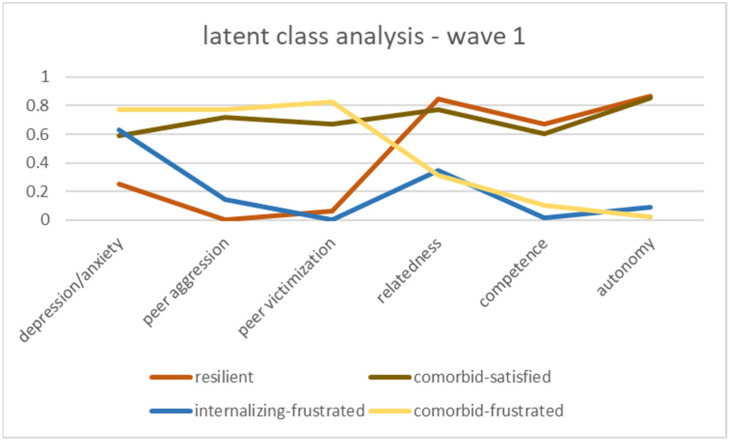
Latent resilience classes, wave 1.

**Figure 2 ijerph-20-05676-f002:**
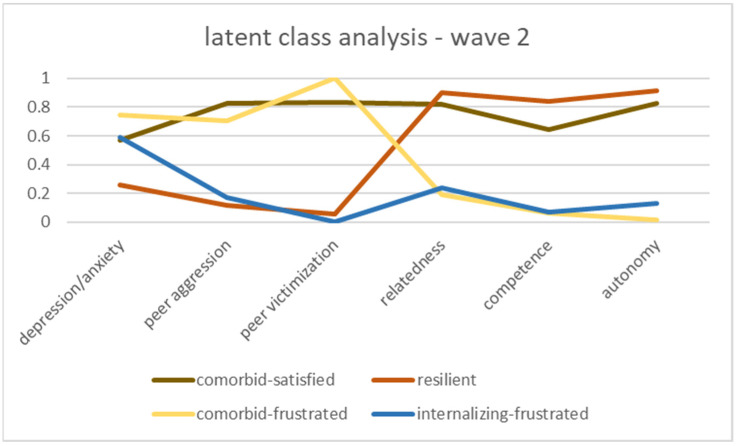
Latent resilience classes, wave 2.

**Figure 3 ijerph-20-05676-f003:**
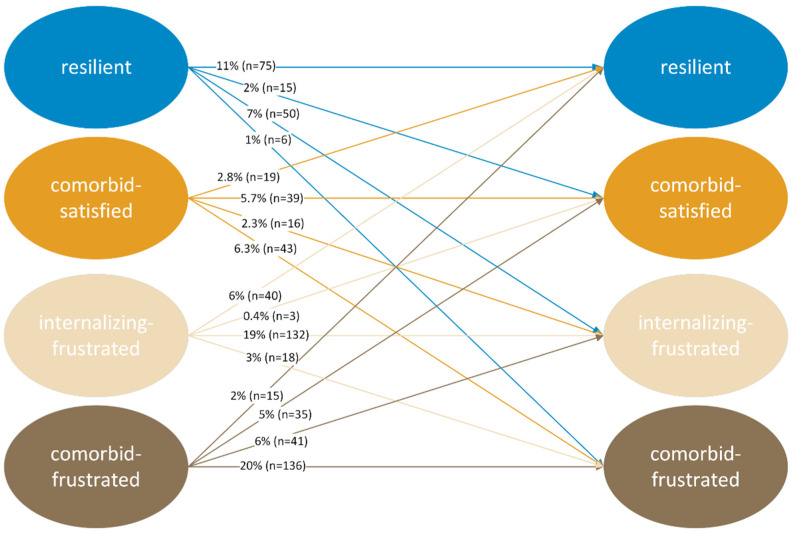
Model-based counts and proportions for each latent transition pattern. Note: We completed listwise deletion due to better model fit, resulting in *n* = 683 for the LTA.

**Table 1 ijerph-20-05676-t001:** Descriptives of indicators and predictors in wave 1 and wave 2.

	Wave 1 (N = 879)		Wave 2 (N = 770)		*t*-Test
Variables	M	*SD*	M	*SD*	Cohens *d/r*
depression/anxiety ^1^	1.585	0.49	1.560	0.50	
peer aggression ^1^	1.501	0.50	1.426	0.49	
peer victimization ^1^	1.479	0.50	1.427	0.49	
relatedness ^1^	1.518	0.50	1.435	0.50	−0.166/0.083 ***
competence ^1^	1.295	0.46	1.300	0.46	0.011/0.005 ***
autonomy ^1^	1.384	0.49	1.352	0.48	−0.066/0.033 ***
gender ^1^	1.477	0.50	1.499	0.50	
socioeconomic status ^3^	1.816	0.69	1.781	0.70	−0.05/0.025 *
migration background ^1^	1.483	0.50	1.503	0.50	0.040/0.020 *
self-esteem ^2^	2.905	0.75	2.931	0.77	0.034/0.017 *
self-efficacy ^2^	2.742	0.59	2.732	0.68	−0.016/0.008 *
social competence ^2^	3.175	0.59	3.049	0.67	−0.200/0.100 *

Note: ^1^ dichotomized; ^2^ scale of 1–4; ^3^ scale of 1–3, with 1 = low, 2 = middle, 3 = high; *** *p* < 0.001, * *p* < 0.05.

**Table 2 ijerph-20-05676-t002:** Latent class analysis model fit for two waves.

Wave	Cla	AIC	BIC	sBIC	Entr	sm%	Cprop	VLMR	LMR	BLRT
1	2	7099.01	7161.96	7120.68	0.75	39.2	0.92 0.95	<0.001	<0.001	<0.001
	3	6950.80	7047.65	6984.13	0.75	21.1	0.92 0.89 0.89	<0.001	<0.001	<0.001
	**4**	**6887.21**	**7017.96**	**6932.21**	**0.75**	**18**	**0.89 0.92 0.89 0.83**	**<0.001**	**<0.001**	**<0.001**
	5	6896.21	7060.86	6952.88	0.77	7	0.88 0.88 0.91 0.79 0.83	>0.05	>0.05	>0.05
2	2	5602.76	5663.16	5621.88	0.81	33.1	0.96 0.93	<0.001	<0.001	<0.001
	3	5434.79	5527.71	5464.20	0.89	30.8	0.94 0.98 0.94	<0.001	<0.001	<0.001
	**4**	**5375.01**	**5500.46**	**5414.72**	**0.85**	**13.9**	**0.79 0.90 0.97 0.95**	**0.018**	0.020	<0.001
	5	5371.32	5529.30	5421.33	0.88	4.1	0.45 0.96 0.90 0.97 0.95	0.005	0.006	0.04

Note: The calculations were made with Listwise = ON; the optimal class solution is bolded; Cla = classes; Entr = Entropy; sm% = smallest group in percentage.

**Table 3 ijerph-20-05676-t003:** LCA models, measurement invariance over time.

Model	df	AIC	BIC	aBIC	∆χ^2^
Fully Unrestricted	63	9678.666	9963.835	9763.802	41.89 (24), 0.01 < *p* < 0.05
Fully Restricted	39	9675.371	9851.905	9728.074	

Note: BIC = Bayesian Information Criterion; sBIC = sample-size-adjusted BIC; AIC = Akaike information criteria. Restricted = item response probabilities held equivalent over time. We completed listwise deletion due to better model fit, resulting in N = 683 for the LTA.

**Table 4 ijerph-20-05676-t004:** Latent transition probabilities of violence resilience classes from wave 1 to wave 2.

	Resilient Wave 2 (*n* = 149)	Comorbid-Frustrated Wave 2 (*n* = 203)	Internalizing-Frustrated Wave 2 (*n* = 239)	Comorbid-Satisfied Wave 2 (*n* = 92)
Resilient wave 1 (*n* = 146)				
	**50.8%**	4.3%	34.1%	10.8%
Comorbid-frustrated wave 1 (*n* = 227)				
	6.8%	**53.4%**	27.1%	15.2%
Internalizing-frustrated wave 1 (*n* = 193)	21.9%	11.4%	**64.8%**	1.9%
Comorbid-satisfied wave 1 (*n* = 117)	17.0%	39.1%	11.1%	**32.8%**

Note: The probabilities of youths staying in their classes from wave 1 to wave 2 are bolded.

**Table 5 ijerph-20-05676-t005:** Multinomial logistic regression analysis: predictors of wave 1 latent resilience classes.

	Internalizing-Frustrated (Ref: Resilient)		Comorbid-Satisfied (Ref: Resilient)		Comorbid-Frustrated (Ref: Resilient)		Comorbid-Frustrated (Ref: Comorbid-Satisfied)		Internalizing-Frustrated (Ref: Comorbid-Satisfied)		Internalizing-Frustrated (Ref: Comorbid-Frustrated)	
Predictor	*B* (SE)	OR	*B* (SE)	OR	*B* (SE)	OR	*B* (SE)	OR	*B* (SE)	OR	*B* (SE)	OR
Female	−0.54 (0.37)	0.58	**0.77 *** (0.32)	2.16	**0.68 *** (0.32)	1.97	−0.10 (0.29)	0.91	**−1.31 ***** (0.33)	0.27	**−1.21 ***** (0.30)	0.30
Low SES	−0.13 (0.26)	0.88	−0.10 (0.24)	0.90	**−0.51 *** (0.24)	0.6	**−0.41 *** (0.19)	0.67	−0.03 (0.20)	0.97	**0.38 *** (0.19)	1.46
No MGB	0.03 (0.33)	1.02	0.38 (0.30)	1.47	0.06 (0.30)	1.06	−0.32 (0.26)	0.72	−0.36 (0.28)	0.70	−0.03 (0.26)	0.97
Self-esteem	**−1.33 ***** (0.35)	0.26	**−1.11 **** (0.34)	0.33	**−1.96 ***** (0.33)	0.14	**−0.85 **** (0.25)	0.43	−0.22 (0.27)	0.80	**0.63 **** (0.30)	1.87
Self-efficacy	**−1.33 ***** (0.37)	0.27	−0.07 (0.32)	0.94	**−1.35 ***** (0.34)	0.26	**−1.29 ***** (0.34)	0.28	**−1.26 ***** (0.35)	0.28	0.03 (0.30)	1.03
Social Competence	**−1.78 ***** (0.41)	0.17	−0.53 (0.43)	0.59	**−1.59 ***** (0.40)	0.21	**−1.06 **** (0.32)	0.35	**−1.25 ***** (0.32)	0.29	−0.19 (0.20)	0.82

Note: *** *p* < 0.001, ** *p* < 0.01, * *p* < 0.05; Ref: reference group, B: Beta coefficient, OR: odds ratio; significant values are bolded.

## Data Availability

The authors will make the raw data supporting this article’s conclusions available without undue reservation.

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
