# Peer review of "Exposure to Intimate-Partner Violence and Resilience Trajectories of Adolescents: A Two-Wave Longitudinal Latent Transition Analysis"

_ijerph, 2023, doi:10.3390/ijerph20095676_

Round 1
Reviewer 1 Report
Thank you very much for the opportunity to review this paper which tackles the important topic of the relationship between exposure to intimate partner violence and resilience trajectories of adolescents. The study employs a novel analysis strategy by conducting a latent transition analysis, which is of particular interest from a life course perspective. As the authors point out, the positive development of youth deserves to be a public health priority, both in terms of building resilience against the onset of mental health challenges which are increasingly prevalent in the adolescent and emerging adult years, and in terms of strengthening the health development of young people before they themselves become parents. Given that parent health, especially mental health, is known to play an important role in the healthy development of children, optimizing all aspects of health in young people who are themselves 'future parents" has the potential for two-generational impact.
The prevalence of psychological IPV in family systems is likely very high, although accurate data are hard to obtain. The sequelae of this may indeed include mental health and behavioral health issues in the young people witnessing what is usually inter-parental IPV, and these behavioral issues often manifest in the school setting. The authors present an excellent and well-written review of the literature on this topic. I am in strong support of the authors conceptualization of resilience as the result of a dynamic transactional process, after the work of Anne Masten. Accordingly, the examination of emergent IPV- exposure-resilience trajectory outcomes is a logical next step in delineating the processes that are operating within the child’s developmental ecosystem (child, family, school, and community) to contribute to the child’s developmental trajectory.
Page 6
Materials and Methods- the authors state that the data come from a longitudinal sample of a broader study of adolescents, using a subsample that have experienced psychological IPV. Could the authors give a little more detail of how representative, or not, the 7th graders in the longitudinal study are of Swiss 7th graders as a whole i.e., was this survey offered only in certain schools and if so, how were the schools chosen? How did the students SES, immigration status etc. relate to population demographics i.e., was this a representative or a skewed sample? Were the reported exposure to IPV rates unusual in any way, or are they likely representative of the population as a whole? Did any students refuse to participate?
Table 1 Page 8. There is a typo- last variable should read “social competence.”
The analysis approach appears sound and the justification for the four-class solution to the latent class analysis is well explained. The latent transition analysis presents novel findings in relation to stability of class membership over time, and across the developmental transition of entry to adolescence. It was of interest and notable that migration background did not predict class membership.
Discussion
This is very well written and explores the results in the light of the literature. The authors make the important point that so many of the adolescents are in the chronic trajectory potentially because ongoing exposure to family IPV is continuing. The impact of the Covid-19 pandemic may of course be relevant here, and the authors discuss this. The discussion as written is clear but is largely an expanded statement of the results. I would have liked the authors to go a little further here from a life course perspective. For example, although we do not know what the family patterns of IPV were earlier in the life course of these young people, it seems at least possible that they were similar to those observed currently…are some of these developmental trajectory pathways being set much earlier in childhood? If so, this has implications for the ideal time to intervene. Waiting for youth to exhibit symptoms in middle school and intervening then would not be optimal from a life course perspective.
In addition, I would have liked more direct suggestions for broader intervention research that could follow from this important work. The authors raise the very important point of the need to raise population awareness of the potentially harmful effects on children of them witnessing both physical and/or psychological IPV in the home- how would the authors suggest that is done? What might be compelling given these data? Should there be an intervention that aims to work with parents to reduce their psychological IPV and to improve the quality of their own relationship? This could have positive health impacts for the parents as well as the child. Focusing on the child’s mental health in this situation and making the child the focus of the intervention, perhaps in the school setting, might not be very effective if the parents relational challenges are left unaddressed. This is especially so if the family members themselves are unaware that there is a problem as is sometimes (perhaps often)the case, especially as patterns of relating often have intergenerational components. Could the authors consider adding just some brief mention of the implications of the findings for life course interventions. Finally, could they consider the potential need of the youth for education about positive relationships and ways of interacting and resolving conflict etc. with the aim of breaking repeated generational cycles of parental discord. This could have a positive impact on the youth’s future life course trajectories and those of their future spouses and children.
Overall, though this is a great paper and I enjoyed reading it. I would strongly suggest the authors consider putting out a companion policy brief based on these findings that lays out more clearly the potential intervention needs, and their potential to improve youth mental health and life course health development.
Author Response
Thank you for your thorough review and insightful comments. We appreciate your suggestions for further clarifying the representativeness of our sample, and provided more detail on the recruitment process, school selection, and demographic characteristics of the students in the revised manuscript [see section 2.1.]. We also corrected the typo.
Revised text in the paper:
The analyzed data come from a longitudinal sample of a broader study of adolescents’ violence resilience conducted in the fall of 2020 (beginning of grade 7, wave 1) and the spring of 2022 (end of grade 8, wave 2). The sub-samples, consisting of adolescents who reported having experienced psychological IPV in the past 12 months, of both waves of the representative convenient sample consisted of 879 (wave 1: 58.3% of the overall sample) and 770 (wave 2: 58.9.% of the overall sample) seventh-grade students from Switzerland. There are currently no statistics available on psychological experiences of IPV in Switzerland, making it difficult to determine the representativeness of reported experiences. This gap in data collection highlights the need for further research and advocacy to shed light on this important issue. However, with the recent adoption of the 2022 motion, it is hopeful that there will be efforts towards the collection of statistical data on psychological IPV experiences in the future [69]. Female (wave 1 sex = 52.6%, wave 2 sex = 54.3%) and male participants (wave 1 sex: 47.4%, wave 2 sex: 45.7%) anonymously completed an approximately 60-minute online questionnaire that research team members introduced verbally on the day of the study in the respective classrooms. As the gender distribution in Switzerland is relatively balanced with 50.4% women, the IPV subsamples contain slightly more women than would be expected in the population [70]. Of the participating students, 51.6% (wave 1) and 51.8% (wave 2) were not Swiss citizens. In Switzerland, the Federal Statistical Office only covers the permanent resident population aged 15 and over, however it is worth noting that 39% of the permanent resident population has a migration background and 52% of children aged 7-14 live in households where at least one parent has a migration background [68]. The participants’ mean age was 11.7 in the first wave (SD = 0.64) and 13.8 (SD = 0.43) in the second. In the first wave of data collection, 37.5% of participants had a low SES, 46.7% a medium SES and 15.6% a high SES. In the second wave, 32.8% had a low SES, 48.8% a medium SES and 15.4% a high SES. While a direct comparison with the population is not available from the Federal Statistical Office, in 2021, 12.6% of the Swiss population had a low educational level, 42.4% had a medium educational level and 45% had a high educational level [71]. Although education level is not equivalent to SES, we can assume that our sample of adolescents with psychological IPV experiences had a lower than average SES, confirming existing studies [72], [73].
The research ethics committee of the University of Teacher Education FHNW approved the project. Once the ethics committee approved the study, we reached out to the cantons of north-western Switzerland, which is a German-speaking area, as well as all secondary school administrations in all four cantons. Subsequently, we contacted all class teachers who then obtained consent forms from parents and students. Participation was voluntary and contingent upon signed declarations of consent and without incentives. This was a non-exhaustive sample, meaning that only students who chose to participate were included in the sample.
Regarding your comments on the discussion section, we agree that the life course perspective is critical in understanding the developmental trajectories of youth exposed to IPV. We therefore elaborated additionally on this perspective and consider the potential implications of our findings for intervention research. We also provided more concrete suggestions for intervention programs that could address the issues you raised [see discussion, paragraphs 2, 3, 8, 9 , 10].
Revised text in the paper:
Overall, our results showed high last-year prevalence rates (about 60% for both waves) of psychological-IPV exposure among adolescents, which is consistent with the findings of Black et al. [21] but at the same time relatively high, possibly because researchers have conducted few studies on the prevalence and incidence of psychological-IPV exposure [20]. We specifically interviewed adolescents and not their parents, possibly leading to under-reporting in the case of parent reports [87], and the rate may have increased during COVID-19 [88]. Regardless of the reasons for the high prevalence, our study highlights the urgent need for representative prevalence and incidence measurements of IPV exposure. The high prevalence and the lack of empirical evidence on psychological IPV clearly indicate that family violence does not receive enough attention, as Sigurdsson [20] points out. Regarding health care institutions, Sigurdsson [20] states that there is a need to remove barriers to reporting violence, raise awareness of characteristics that indicate possible exposure to IPV, and promote education about IPV. As adolescents spend most of their time at school, we believe that raising awareness of possible exposure to violence can help to sensitize school staff and demonstrate the importance of reporting violence.
To address and improve prevalence rates of family violence, it is necessary for various stakeholders to be involved. Caregivers, educators, and professionals should understand the negative impact of stigma and shame on children and adolescents who have experienced parental violence, as this affects their reporting [89]. Furthermore, governments have a crucial role to play in preventing and responding to family violence through policies, legislation and funding. They can develop and implement programs and services that support victims, raise awareness and hold perpetrators accountable through the explicit prohibition of family violence, which is still lacking in Switzerland [69].
(....)
Providing early mental health identification programs in school settings, such as universal screening and comprehensive clinical assessments, can improve mental health care for children and adolescents in need and their academic performance [101]. In this scope, teacher training and support to identify and adequately refer children and adolescents [102] and parent training concerning awareness, early identification, and intervention needs [6] are fundamental. Mental health literacy programs for the various actors involved can also play an important role in increasing mental health knowledge and promoting positive attitudes toward students’ mental illness [103], reduce stigma-related attitudes and behaviors, and increase teachers’ confidence in helping students [104], [105]. As early and enduring exposure to IPV can cause harm to children's brain development and behavior [106], [107], the negative effects may not be immediately evident, leading to delayed effects that can emerge from months to years after the exposure [108]. Given this knowledge, there is a strong argument for early intervention to mitigate the immediate impacts of IPV exposure and prevent future problems [106].
However, there are still many open questions about the relationship between violence exposure and its impact on adolescence, particularly regarding the appropriate time and method to intervene in order to promote well-being [109]. Some studies claim that there is insufficient evidence about the positive impact of interventions due to methodological issues [110], [111], while others suggest that interventions such as parent-focused IPV programs or interventions with mothers and children jointly or separately can mitigate or even withdraw the negative impacts of IPV exposure [90], [107], [109], [112]. It is important to note that universal interventions may not be sufficient for adolescents who have experienced certain forms of violence, such as psychological IPV [113]. Previous interventions for women who have experienced IPV are often non-specific and not theoretically based, which can lead to poor outcomes. Children and adolescents are even less likely to be included in these interventions, highlighting the need for more research and tailored interventions for this population [114]. To address the underlying factors of IPV, such as gender inequality or socioeconomic deprivation, some authors argue for the promotion of policies and the coordination of targeted services within the individual’s context (e.g., home, school, community) [115]. For example, at the school level, interventions tailored to adolescent’s needs that include both trauma-specific and non-trauma-specific content, as well as the creation of a positive school climate and safer school environments, are recommended [107]. Some reviews suggest the importance of early intervention through universal programs and the use of an adaptation-based approach to resilience in complement with a strengths-based approach to promote positive development for children and youth growing in adversity [109], [116], [117].
As a global burden, family violence is a significant public health problem that has a destabilizing effect on the normative development of future generations and thus fosters further violence, inequality and misery. Future research should therefore focus on developing a more comprehensive understanding of the various factors that contribute to the resilience of young people, their families and society, by examining in depth the extent to which young people experience violence and its consequences. Even if schools are an appropriate place to educate young people about violent behavior and experiences, the first step is to train teachers so that they have the positive attitudes, materials and skills to bring about these changes. In addition, there is a need to explore the concept of family resilience, which has received relatively little attention in resilience research, and which will be instrumental in reducing the prevalence of family violence, mitigating its negative effects and improving the resilience of victims and their families.
Finally, we appreciate your suggestion of creating a companion policy brief based on our findings, and we will consider this as the next step in disseminating our research to a broader audience.
Thank you again for your valuable feedback and for taking the time to review our manuscript.
Reviewer 2 Report
Overall, in my view this is a well-written paper, with substantial data analysis. However, my main concern for the paper is the author assumed the incidence rate of exposure to psychological IPV, without baseline data. Although the authors mentioned this as the limitation of the paper, I don't think the authors can tell this is incidence rate as such, rather as past-year IPV exposure. With this concern, I am not sure if the analysis employed in the paper is entirely correct, and how the author can preclude the effect of lifetime IPV if any.
Author Response
Thank you for your feedback and for pointing out the error in using the term "incidence" instead of "prevalence". We mistakenly used the term "incidence" when we actually meant "prevalence", due to the use of the terms "incidence" and "prevalence" frequently ambiguously applied in research.
We apologize for any confusion this may have caused. We appreciate your attention to detail and the opportunity to clarify this terminology. We changed the term incidence rate to prevalence rate. Regarding your concern about assuming the incidence rate of exposure to psychological IPV without baseline data, we acknowledge that our study focused on past-year IPV exposure. This is also mentioned in the limitation section. Our study was designed to investigate the current prevalence and associated factors of psychological IPV exposure, rather than to assess the incidence of IPV exposure. We believe that this is an important area for future research.
Thank you again for your valuable feedback and for taking the time to review our manuscript.
Reviewer 3 Report
Well written. Such an important topic that is understudied.
Clarify sentence line 48-50, e.g. Violence resilient refers to those who develop adaptively despite...
When defining resilience, the authors state that resilience as a trait has been criticized because it was interpreted as victim blaming. In my own research and clinical practice with adult persons, identifying as women, living houseless, resilience was assumed to be present but was unrecognized by the individual and suppressed by the extreme living circumstances (sociodemographic, cultural, and economic structure as factors). This may suggest that examining resilient factors as a process can inform the types of interventions needed. Evaluating resilience as an outcome is mechanistic and suggests some type of finality, when we know that the results of IPV can be protracted, as is well documented in the authors' manuscript. The definition of the resilience trajectory does not seem to support viewing resilience as an outcome. Section 1.3 suggests that behaviors are resilient outcome indicators. Might they be IPV exposure outcome indicators? The study results also suggest process.
Author Response
Thank you very much for your valuable input. Your perspective on resilience as a process rather than a fixed trait is insightful and aligns greatly with our own view. We appreciate your feedback and have made efforts to clarify our use of the term "resilience outcomes" in the paper [see Introduction, pararaph 2].
Edited text in the paper:
Research has clearly indicated that a major risk factor for healthy development that youths face is exposure to family violence. Youths are at significant risk of lifelong negative consequences that are also reflected in society [7], [8], with intimate-partner violence (IPV) exposure being one of the most common familial burdens for youths [9]. Violence resilience refers to a process where individuals who develop adaptively despite experiences of violence that lead to a severely increased likelihood of negative socio-emotional consequences. [10]. According to Masten [11], resilient outcomes are characterized by competence at age-specific developmental tasks as well as an absence of psychopathological symptoms despite risk exposure. Although resilience is seen as a process, in order to methodically capture a moment in time or a trajectory of resilience, resilience outcomes are measured. When defining resilience outcomes, we always assume that they are a snapshot of a process and not a final outcome. The present study draws on Deci and Ryan’s [12] self-determination theory, which provides extensive evidence that meeting basic psychological needs (autonomy, relatedness, competence) increases people’s abilities and adaptability and reduces their vulnerability to psychopathology [13]. Psychopathological behaviors stemming from IPV exposure that have been particularly well studied include depressive [14] and aggressive behaviors [15] along with the lesser studied but equally important peer victimization [16].
We agree that resilience is a dynamic process that can change over time, is influenced by various factors, and that capturing specific moments in this process is an important aspect of understanding resilience. We have made sure to highlight this perspective in the text to avoid any confusion or misinterpretation. We also appreciate your emphasis on the importance of considering the broader context and social factors that may impact individuals' resilience and overall well-being, particularly in the case of youth who have experienced violence. We ensured to emphasize the need for a comprehensive and nuanced approach to understanding resilience in our research. Thank you for your suggestion regarding the title of section 1.3, which we have now revised to better reflect the content.
Edited title in the paper:
Domains for non-normative development in the face of psychological IPV
Thank you again for your valuable feedback and for taking the time to review our manuscript.
Round 2
Reviewer 2 Report
I am happy with the modification provided by the authors.